# Assessing the Impact of the COVID-19 Pandemic on Pregnant Women’s Attitudes towards Childhood Vaccinations: A Cross-Sectional Study

**DOI:** 10.3390/vaccines12050473

**Published:** 2024-04-29

**Authors:** Paola Arcaro, Lorenza Nachira, Fabio Pattavina, Enrica Campo, Rossella Mancini, Domenico Pascucci, Gianfranco Damiani, Brigida Carducci, Antonietta Spadea, Antonio Lanzone, Stefania Bruno, Patrizia Laurenti

**Affiliations:** 1Section of Hygiene, University Department of Life Sciences and Public Health, Università Cattolica del Sacro Cuore, 00168 Rome, Italy; paola.arcaro01@icatt.it (P.A.); lorenza.nachira01@icatt.it (L.N.); enrica.campo01@icatt.it (E.C.); rossella_mancini@msn.com (R.M.); gianfranco.damiani@unicatt.it (G.D.); stefania.bruno@unicatt.it (S.B.); patrizia.laurenti@unicatt.it (P.L.); 2Women, Children and Public Health Sciences Department, Fondazione Policlinico Universitario A. Gemelli IRCCS, 00168 Rome, Italy; brigida.carducci@policlinicogemelli.it (B.C.); antonio.lanzone@unicatt.it (A.L.); 3Health Management, Fondazione Policlinico Universitario A. Gemelli IRCCS, 00168 Rome, Italy; 4Local Health Authority, ASL ROMA 1, 00193 Rome, Italy; antonietta.spadea@aslroma1.it

**Keywords:** vaccination, COVID-19, pandemic, pregnant women, attitudes

## Abstract

The COVID-19 pandemic has globally disrupted immunisation practices, impacting vulnerable populations such as pregnant women (PW), who harbour concerns about future children’s immunisations. This study aimed to assess the pandemic’s impact on PW’s attitudes towards childhood vaccinations. During three consecutive flu seasons from October 2019 to January 2022, a cross-sectional study was conducted in a large Italian teaching hospital using a questionnaire. The chi-square test was performed to compare each season. Across the 2019–2020 to 2021–2022 seasons, course attendance by PW surged from 105 to 340. Significant shifts in vaccination intentions were noted, including a 7.5% decrease in measles vaccination intent (*p* = 0.02) and a 10% decrease in that of pertussis (*p* = 0.004) from 2019–2020 to 2020–2021. While perceived contagion risk decreased, disease severity perceptions increased, with few significant differences. A statistically significant reduction was noted in the proportion of participants suspecting economic motives behind NHS workers’ promotion of childhood vaccinations. Furthermore, the pandemic period saw an increase in the perceived utility of non-institutional websites and the advice of physicians outside the NHS. These findings will help develop evidence-based, tailored interventions and communication strategies to address vaccine hesitancy and ensure optimal vaccination coverage among children born during and after the pandemic.

## 1. Introduction

Vaccination is the most extensively recognised factor that has decreased the occurrence of severe infectious diseases that were once widespread [1]. Globally, more than 2.5 million child deaths can be prevented each year thanks to vaccinations, with the potential to prevent a further 2 million deaths among children with increased access to vaccines in developing countries [2].

According to the World Health Organization (WHO), unfortunately, the COVID-19 pandemic has caused a decrease in the overall vaccination rate for most vaccinations by about one or two percentage points, leading to a decrease in global coverage, which dropped from 86% in 2019 to 81% in 2021 [3]. As a consequence of this phenomenon, at least 80 million children under the age of one are at risk of diseases such as diphtheria, measles, and polio, as COVID-19 has disrupted routine vaccination efforts [4].

In light of this, it is crucial to direct special attention towards risk groups, particularly pregnant women (PW). Vaccinating PW is indeed a vital aspect of their medical care [5], as both they and their children are at a higher risk of severe illness and complications from infectious diseases.

As regards vaccinations in PW, Centers for Disease Control and Prevention (CDC) recommend one dose of the influenza vaccine, administered at any time during pregnancy, and a dose of the tetanus, diphtheria, and pertussis (Tdap) vaccine, between 27 and 36 weeks of gestation, in each pregnancy [6].

During 2019–2020, in the United States, 61.2% of PW received influenza vaccination, 56.6% received Tdap during pregnancy, and 40.3% received both vaccines [7]. In Italy, the uptake of the influenza and pertussis vaccines among PW for the influenza season 2018–2019 was 14.9% and 60.9%, respectively, with only 13% receiving both [8]. Another study conducted in Southern Italy between October 2021 and April 2022 showed that only 21.1% and 36.5% of women received influenza and Tdap vaccines, respectively, during pregnancy [9]. The suboptimal coverage of maternal vaccination (estimated at 0–70%) with seasonal influenza and pertussis vaccines worldwide represents a missed opportunity to improve the health of mothers and newborns [10,11,12].

Administering influenza and pertussis vaccines to PW is now a standard and safe practice [13]. However, the success of vaccines relies not only on their effectiveness and cost-effectiveness but also on their acceptance by the population [14]. In addition, the perceived safety of vaccination is directly related to the perceived trust or distrust of the health system [15].

One of the main factors contributing to low vaccination coverage has been identified as vaccine hesitancy, which is represented by doubts due to a lack of knowledge or other influencing factors [16]. This trend has become more evident because of the COVID-19 pandemic, which led to a disturbance in the administration of immunisation campaigns due to the efforts of healthcare workers in maintaining operational services and the increased public anxiety regarding access to healthcare facilities, as well as restrictions on movement [17]. Moreover, health communication activities and informative sources could have had a significant impact on health-related attitudes, beliefs, and behaviours.

The aim of this study is to assess whether the pandemic has led to changes in vaccination attitudes among PW attending antenatal classes that took place at the Fondazione Policlinico Universitario A. Gemelli (FPG), a large research hospital in Rome. Specifically, the aim was to assess the potential impact of the pandemic on their perceived usefulness of vaccination information sources, their trust in healthcare workers (HCWs) and the National Health Service (NHS), their perception of the risk of infection and severity of vaccine-preventable diseases (VPDs), and their vaccination intentions.

## 2. Materials and Methods

### 2.1. Study Design and Timeframe

A repeated cross-sectional study was conducted at the FPG across three flu epidemic seasons, each time involving different groups of participants: one before the COVID-19 pandemic, running from October 2019 to January 2020, and two during the pandemic, from September 2020 to January 2021 (the second wave of the COVID-19 pandemic) and from October 2021 to January 2022 (the fourth wave of the COVID-19 pandemic), respectively [18]. The study timeframe was consistent with Italian recommendations regarding flu vaccination [19,20,21]. The methodology used is in accordance with the most recent Guidelines for Observational Studies, STROBE (Strengthening the Reporting of Observational Studies in Epidemiology) [22].

### 2.2. Study Sample and Setting

The study involved convenience samples, represented by PW attending the antenatal classes that took place at the FPG during the three above mentioned periods. The courses were organised by the Obstetric and High-Risk Pregnancy Unit within the Department of Women’s, Children’s Health, and Public Health. In accordance with FPG corporate policy, these educational sessions began in the fourth month of gestation, targeting both expectant mothers and their partners. Designed to be congruent with the progression towards childbirth, the courses do not stipulate specific inclusion criteria, thereby ensuring accessibility to all pregnant women followed by the FPG hospital. All women who attended the courses and could provide written informed consent (for the first course) or online informed consent (for the second and third courses) were included.

### 2.3. Questionnaire and Data Collection

For the purposes of this study, we asked women who attended the antenatal classes to answer an anonymous questionnaire about their knowledge and beliefs towards vaccination in general, their knowledge and attitudes regarding childhood vaccinations, which are mandatory or recommended in Italy, and their trust in institutions and healthcare workers (HCWs). The questionnaire was previously validated in a multi-centric Italian study [23,24].

The intention to vaccinate children against VPDs for which vaccination is mandatory or recommended in Italy was assessed with a multiple-choice question. Trust in HCWs and the NHS and perceptions of the risk of infection and severity of VPDs were assessed using linear scale questions with a 4-point scale ranging from “not at all” to “very much”. The usefulness of various sources of information was rated on a scale from 1 (not useful at all) to 5 (very useful).

Due to the outbreak of the SARS-CoV-2 pandemic and the subsequent declaration of a state of emergency by the Italian government in January 2020 [25], there were substantial differences in data collection methods in the three seasons. In 2019–2020, the course was held on site at the FPG, and questionnaires were handed out to participants. During 2020–2021 and 2021–2022, the course was delivered through an online meeting platform, and questionnaires were administered as online forms.

### 2.4. Data Analysis

The intention to vaccinate children against VPDs was expressed as the percentage of respondents who selected each vaccination from the list. Trust in HCWs and the NHS and the perception of the risk of infection and severity of VPDs were expressed as the percentage of respondents who answered “quite” or “very”. For the usefulness of different information sources, the mean scores for each source were calculated, and the sources were ranked accordingly. To assess differences between the three periods (2019 vs. 2020, 2019 vs. 2021, and 2020 vs. 2021), a chi-square test was performed, setting statistical significance at *p* = 0.05. All statistical analyses were carried out using the software “Stata 16” (Stata Corp, Lakeway, TX, USA).

### 2.5. Ethical Statement

This study is compliant with the Local Ethical Committee Standards of the FPG. It was approved, registered (Prot. N° 38264/19 ID: 2782), and carried out in accordance with the Helsinki Declaration and EU Regulation 2016/679 (GDPR).

## 3. Results

### 3.1. Socio-Demographic and Clinical Characteristics

A total of 105 PW attended the course in the 2019–2020 season, 317 in the 2020–2021 season, and 340 in the 2021–2022 season. Of these, 104 (99%), 241 (76%), and 160 (47%) completed the questionnaire during the corresponding periods. Table 1 presents data on participants’ citizenship, marital status, education, employment, and trimester of pregnancy for each season. Most participants were Italian citizens, married, and university graduates; the mean ages were 34.5 (standard deviation (SD) = 4.9), 33.8 (SD = 4.0), and 35.2 (SD = 4.4), respectively. The characteristics of the participants were similar across all three periods, except for participant age, which presented a statistically significant difference (*p* = 0.001) between the 2020–2021 and 2021–2022 seasons.

### 3.2. VPDs: Intention to Vaccinate, Perceived Risk and Perceived Severity

Table 2 shows the percentages of participants who intended to vaccinate their children with compulsory and recommended vaccinations during the three different periods considered; data for hepatitis B was missing for the 2020–2021 season. Statistically significant changes were observed over the years: percentages for hepatitis B, pertussis, and measles decreased, while those for *Haemophilus influenzae b*, diphtheria, HPV, and meningitis increased.

Table 3 shows the percentage of participants who believed it was quite or very likely that their children could contract the surveyed VPDs. There were statistically significant decreases in the percentages observed for hepatitis B, *Haemophilus influenzae b*, rubella, HPV, and meningitis during the pandemic periods compared to the pre-pandemic period.

Table 4 shows the percentage of participants who considered the surveyed VPDs quite or very severe. There was a statistically significant increase in the percentages observed for diphtheria and HPV, and a statistically significant decrease for measles and meningitis.

### 3.3. Trust in Healthcare Workers and the National Health Service (NHS)

Table 5 shows the changes in the respondents’ confidence in HCWs and the NHS over the three seasons considered.

The percentage of respondents who reported having confidence in information they received from HCWs ranged from 94.9% to 98.1% throughout the three flu seasons, without any statistically significant change. Additionally, between 92.1% and 95.1% of participants agreed that NHS workers are prepared and updated on vaccinations, with no statistically significant variations. Statistically significant decreases were observed in the percentage of participants who believed NHS workers had economic interests in promoting childhood vaccinations.

The percentage of respondents who reported having more trust in providers outside the NHS increased from the first to the second season, while showing a statistically significant decrease in the third season. The percentage of respondents who believed that NHS operators give information only about the benefits of vaccines and not the risks decreased significantly between pre-pandemic and pandemic periods.

### 3.4. Perception of the Usefulness of Different Information Sources

Table 6 shows the ranking of the perceived usefulness of information sources during the three seasons. Participants relied primarily on gynaecologists, antenatal classes, and institutional websites for their vaccination information, followed by word of mouth/friends/acquaintances.

The score regarding the perceived usefulness of non-institutional websites, vaccination clinics, and physicians outside the NHS increased during the pandemic, while that of Local Health Authority/Ministry of Health information brochures, mass media, and general practitioners lost value as information sources decreased.

## 4. Discussion

This study aimed to investigate the impact of the pandemic on the attitudes and beliefs of pregnant women towards routine vaccinations during childhood and adolescence over three periods. The results showed some differences between the pre-pandemic and pandemic periods concerning women’s intentions to vaccinate their children for the studied infectious diseases, as well as improvements in their trust in healthcare workers and the healthcare system. The study also highlighted a general decline in the perception of the risk of contagion and an increase in the perception of some diseases’ severity, even if few statistically significant differences were found. Lastly, in terms of the perceived usefulness of the information sources studied, no major changes were noted in how they ranked.

### 4.1. Vaccination Intention

The results generally indicated a strong inclination to vaccinate children against the VPDs, with percentages consistently surpassing 75% across all three periods. On the same topic, a systematic review conducted by Whang et al. reported a prevalence of parental willingness to childhood vaccination of 47.3% during the pandemic period [26]. However, since socio-economic factors have a positive influence on intention to vaccinate children [26], the high level of education of our sample should be considered, as approximately 80% of the women had a university degree. The only exceptions to the high vaccination intention recorded were observed for *Haemophilus influenzae type b* and *Human Papillomavirus* (HPV), which reported percentages between 50 and 60%. In the case of *Haemophilus influenzae b*, the disease may be scarcely known, and it could be confused with the influenza virus, which is generally perceived as low-risk for children and adolescents, and the recommended vaccine is considered not very effective [27]. Regarding HPV, as the vaccine is recommended from the age of 11 and the disease is primarily transmitted through sexual contact, it’s probable that the perceived urgency to vaccinate children is delayed. To support our findings and considerations, a study by Helmkamp et al. showed a higher hesitancy towards influenza and HPV vaccines than for childhood vaccines (*p* < 0.0001 for both flu and HPV vaccines), with 6.8% of parents hesitant towards childhood vaccines, 26.1% towards the influenza vaccine, and 25.6% towards the HPV vaccine [27]. Gencer et al. reported a positive effect of the pandemic on pregnant women’s decisions to vaccinate their children in the future, with 50% of them reporting a positive effect, compared to 8.6% who reported negative effects, and 41.4% who reported no change [28]. Furthermore, scientific literature reported that most parents considered routine vaccinations on schedule important for their children during the pandemic [29,30]. Except for pertussis, measles, and hepatitis B vaccinations, our results are consistent with these findings, suggesting the pandemic does not seem to have adversely affected the intention to vaccinate. However, we observed a general decrease in vaccination intentions during the first year of the pandemic, followed by an increase in the second. Only HPV and meningitis vaccinations showed a continuous upward trend over time, with statistically significant results recorded in the second year of the pandemic, suggesting a positive response to the campaigns conducted to promote these recommended vaccines in Italy. In particular, meningococcal vaccination, along with pertussis and measles, recorded the highest percentages of intention to vaccinate, in agreement with a recent survey conducted in several countries by Tan et al. [29]. According to this study, most parents considered vaccinations against measles, meningitis, and pertussis to be very important for their children, with percentages reaching 95%, 94%, and 92%, respectively. Furthermore, the percentage of parents deeming meningitis vaccination important reached 97% in Italy. Nevertheless, although the pandemic may have certainly had an impact on overall attitudes towards vaccination [29,31], the findings on vaccination intention suggest that the low vaccination coverage observed worldwide during the pandemic is primarily attributable to disruptions in routine immunisation services [32,33], rather than to a genuine decline in the propensity for vaccination.

### 4.2. Perception of the Risk of Disease Contagion

Except for poliomyelitis and diphtheria, our findings showed a general decrease in the percentage of women who considered the risk of infection to be high, with some results being statistically significant. The highest percentages were found for mumps, chickenpox, measles, and rubella—commonly known to be highly contagious and for which a tetravalent vaccine is compulsory in Italy. Unfortunately, there is insufficient literature on this topic for comparative analysis. Nevertheless, it is important to consider the significant impact of the pandemic on the epidemiology of most infectious diseases [34,35], due to the stringent measures and lifestyle changes that occurred during that period. Facchin et al. argued that the sudden decrease in measles incidence observed during the pandemic period in Italy was most likely attributed to the non-pharmacological measures implemented to prevent the transmission of SARS-CoV-2 [36]. As a result, the pandemic’s impact on the decrease in disease incidence may have affected the perception of the risk of contagion.

### 4.3. Perception of Disease Severity

Exploring the perception of disease severity is crucial, as it is one of the six constructs of the Health Belief Model (HBM), which suggests that individuals are more likely to adopt preventive health behaviours, like vaccination, when they perceive the disease as a significant threat, believe in the vaccine’s effectiveness, and perceive minimal barriers to obtaining it, along with external cues and confidence in one’s ability to act [37,38]. Our findings revealed that during the pandemic, there was a nearly uniform increase in the percentage of women who perceived the severity of diseases, with few statistically significant differences. A concerning exception pertained to measles. Despite the heightened perception of its contagiousness (albeit decreasing during the pandemic period), the study revealed a simultaneous decline in vaccine intention and a significant decrease in the perception of disease severity in the second year of the pandemic. This poses a potential public health hazard, considering the possible repercussions on vaccination coverage. Another notable exception is observed for meningitis, which showed a significant reduction in perceived severity over time despite having the highest recorded percentages among the diseases studied. Nevertheless, the perception of the severity of infectious disease during the pandemic may be mainly ascribed to emotional distress, maladaptive behaviours, and other indirect discomforts induced or exacerbated by COVID-19. These factors could indeed interact collectively to shape public perceptions, contributing to an amplified sense of the seriousness of all illnesses and diseases [39,40].

### 4.4. Trust in Healthcare Workers and the National Health Service

Ozawa et al. [41] argued that during systemic shocks like pandemics, mistrust is reinforced through reciprocal interactions between health and immunisation systems, persisting beyond system restoration and extending to the broader health system. Moreover, Stolzenberg et al. found that both the levels of perceived trustworthiness and respectability of healthcare workers decreased following the pandemic [42]. Our results partially confirmed these findings. Despite over 90% of women expressing confidence in the information provided by healthcare providers and perceiving NHS workers as adequately prepared and updated about vaccinations, a slight decline in confidence was recorded during the pandemic. Simultaneously, an increase in trust towards other providers outside the NHS was highlighted, although the results were not statistically significant. In the initial phase of the pandemic, uncertain communication from the scientific community through the media may have contributed to shifting trust to other providers. Conversely, during the pandemic period, there was a decrease in the perception that NHS workers have economic interests in vaccinations, as well as in the belief that they only communicate the benefits of vaccination without addressing its risks. However, almost a quarter of the women during the second year of the pandemic reported that the information they received was partial, focusing only on the benefits of vaccines. These findings are relevant, considering that Gualano et al. demonstrated that individuals who believe healthcare professionals have economic interests in child immunisation and provide information only on vaccination benefits are less likely to support compulsory vaccination (OR: 0.66, CI 95%: 0.46–0.96, *p* = 0.03; OR: 0.66, CI 95%: 0.46–0.95, *p* = 0.03, respectively) [23].

### 4.5. Sources of Information

Findings revealed that gynaecologists, institutional websites, and antenatal classes are perceived as the most useful over time, with the latter dropping in rank from first to third position in the 2021–22 period. Although gynaecologists and professionals in antenatal classes are not primarily involved in childhood vaccinations, they likely represent the reference figures for these women during pregnancy. In this regard, Vogels-Broeke et al. showed that more than 80% of pregnant women found information from midwives, obstetricians, and antenatal classes useful [43]. Another study conducted in Australia found that women considered midwives and childbirth education classes the most useful sources of information during pregnancy, with only 12.8% relying on obstetricians [44]. Nevertheless, several studies support the idea that trust in institutional and professional information sources, as well as their utilisation, positively influences vaccination intention and, consequently, vaccine acceptance [45,46,47]. In this context, Charron et al. reported that vaccine acceptance was higher when parents received information from healthcare professionals rather than from the internet or relatives, and the rate of vaccine hesitancy was higher among parents who had obtained information from all three of these sources (70.9%) (OR = 4.6; *p* < 0.0001) compared to those who had received information only from healthcare professionals (34.6%) [48]. Worrisome findings include word of mouth being ranked fourth over the years, and non-institutional websites moving from eighth to fifth during the pandemic period. These findings are also consistent with those reported in other Italian studies during the pre-pandemic period [23,49,50] and represent a noteworthy trend, since scientific literature recognises these sources as primary contributors to misinformation [51,52,53,54]. Moreover, a study conducted in Saudi Arabia reported that seeking vaccination information from family, friends, or social media increased delayed vaccinations [55]. The results indicate that the overall perceived usefulness of the internet, whether specifically referring to institutional or non-institutional websites or to mobile applications, is supported by previous studies that recognise it as one of the primary sources of information [56,57]. In this regard, it must be considered that an appropriate use of this kind of media could positively impact health behaviour, including vaccine uptake [58]. Moreover, general practitioners are towards the bottom of the ranking across all three periods; conversely, My et al. [59] identified them as the most influential and the primary information source for 83% of parents, followed by government or health authorities (28%) and the internet (27%).

At the macro level, aimed at national and regional policymakers, the focus should be on strengthening public health campaigns to emphasise the importance of vaccinations for diseases like pertussis, measles, and hepatitis B. The promotion of accurate information from institutional and official sources is crucial to countering vaccine misinformation, thus highlighting the role of health authorities in promoting public trust in vaccination efforts. At the meso level, related to the local components of the National Health Service, the priority is ensuring vaccine accessibility. This includes minimising disruptions in routine immunisation services during health emergencies to sustain or improve vaccination rates, showcasing the critical function of local healthcare systems in maintaining vaccine access. At the micro level, efforts focus on education and trust-building within communities and between healthcare providers and patients. Training healthcare workers on vaccination guidelines and communication, enhancing provider-patient relationships through transparency, and emphasising the importance of credible information sources are key. Encouraging vaccination discussions, especially during prenatal consultations, and empowering pregnant women with knowledge, alongside leveraging healthcare networks to fight misinformation, are vital for fostering vaccine acceptance and coverage.

### 4.6. Strengths and Limitations

The findings of this study must be interpreted considering both its limitations and strengths. A limitation is the use of convenience sampling, which hampers the ability to generalise our results to all pregnant women. In particular, those who attend antenatal classes might be more health-conscious about themselves and their children than those who do not participate in such courses. Furthermore, the transition to online courses and surveys during the COVID-19 pandemic may have affected response rates and participant engagement, also influenced by changes in emotional well-being and priorities. Yet, it is noteworthy that this research is pioneering in exploring the effects of the pandemic on the attitudes and beliefs of pregnant women towards routine vaccinations for children and adolescents. Another weakness is the reliance on self-reported data, which carries a risk of recall bias. Moreover, the requirement for internet access for participant selection introduces a potential for selection bias. The subjective nature of the data collection also raises concerns about the inclination of participants to provide socially desirable responses, thus possibly skewing the results. Nonetheless, the statistical analyses were rigorously performed within established methodological frameworks, drawing on the existing body of scientific literature, which strengthens the validity of the conclusions. Lastly, this study addressed only some of the factors influencing attitudes towards childhood vaccines and their uptake. This allows for the development of further studies focused on the nature and degree of the interplay between these factors before and after the pandemic. Future research should delve into how socio-economic factors influence vaccination intentions and seek to understand the specific reasons behind hesitancy towards certain vaccines. Additionally, it’s critical to assess how non-institutional information sources contribute to misinformation. Continued monitoring of vaccination intentions, behaviours, and trust in the healthcare system is essential for identifying trends and pinpointing areas that require intervention. This approach will not only deepen our understanding of these dynamics but also inform strategies to promote vaccination among pregnant women in the context of evolving public health challenges.

## 5. Conclusions

Our findings suggest that PW’s attitudes and beliefs towards childhood vaccinations might have been modified by the COVID-19 pandemic, in particular regarding risk perception and trust in NHS healthcare workers. Other studies, carried out on the Italian PW population are needed to provide evidence for policymakers, health service planners, and the broader public health community. This will ensure the timely implementation of strategies and interventions to maintain vaccine confidence and mitigate potential adverse public health effects.

## Figures and Tables

**Table 1 vaccines-12-00473-t001:** Socio-demographic and clinical characteristics of pregnant women who answered the questionnaire in the three flu seasons.

Variables	2019–2020 (*n* = 104) %	2020–2021 (*n* = 241) %	2021–2022 (*n* = 160) %	*p*-Value 2019–2020 vs. 2020–2021	*p*-Value 2019–2020 vs. 2021–2022	*p*-Value 2020–2021 vs. 2021–2022
**Demographic and educational**						
Italian citizenship	95.2	97.5	96.9	0.25	0.23	0.65
Married	99.0	99.2	96.8	0.91	0.12	0.35
Graduate	77.9	79.2	81.9	0.82	0.74	0.78
**Pregnancy characteristics**						
First Pregnancy	96.2	92.5	92.5	0.20	0.11	0.50
Third Trimester	86.5	90.9	91.2	0.22	0.88	0.55
**Age**	34.5 (4.9) *	33.8 (4.0) *	35.2 (4.4) *			

* Age values are expressed as “mean (standard deviation)”.

**Table 2 vaccines-12-00473-t002:** Percentage of pregnant women who declared the intention to vaccinate their children for compulsory and recommended vaccination before and during the COVID-19 pandemic in the 2019–2020, 2020–2021, and 2021–2022 flu seasons.

Claims	2019–2020 %	2020–2021 %	2021–2022 %	*p*-Value 2019–2020 vs. 2020–2021	*p*-Value 2019–2020 vs. 2021–2022	*p*-Value 2020–2021 vs. 2021–2022
Hepatitis B	87.5	n.a. *	86.3	-	**0.04**	-
Poliomyelitis	76.0	78.8	82.5	0.55	0.27	0.81
*Haemophilus influenzae b*	58.7	58.5	59.4	0.98	**0.005**	0.56
Tetanus	80.8	80.9	85.6	0.97	0.14	0.89
Diphtheria	77.9	76.4	83.4	0.75	0.36	**0.04**
Pertussis	95.2	85.5	90.0	**0.004**	0.10	0.09
Measles	94.2	86.7	88.1	**0.02**	0.18	0.66
Rubella	85.6	83.8	86.9	0.34	0.09	0.21
Mumps	80.8	80.1	84.4	0.44	0.26	0.86
Chickenpox	83.7	76.4	83.8	0.06	0.12	0.96
Human Papillomavirus (HPV) infection	51.0	53.1	57.5	0.64	**0.03**	0.80
Meningitis	87.5	87.6	90.0	0.50	**0.03**	0.77

* n.a. = not available. Statistically significant results are in **bold**.

**Table 3 vaccines-12-00473-t003:** Percentage of pregnant women who considered their children’s contagion probable (quite-very probable) with regard to the following vaccine preventable diseases.

Claims	2019–2020 %	2020–2021 %	2021–2022 %	*p*-Value 2019–2020 vs. 2020–2021	*p*-Value 2019–2020 vs. 2021–2022	*p*-Value 2020–2021 vs. 2021–2022
Hepatitis B	48.4	41.8	43.2	**0.05**	**0.04**	0.38
Poliomyelitis	21.1	23.9	23.5	0.51	0.27	0.21
*Haemophilus influenzae b* infection	71.1	63.5	56.6	**0.02**	**0.005**	0.16
Tetanus	66.7	57.4	54.3	0.18	0.13	0.38
Diphtheria	35.6	38.1	36.3	0.44	0.36	0.39
Pertussis	79.8	71.8	65.6	0.12	0.10	0.36
Measles	89.6	86.1	85.7	0.06	0.18	0.74
Rubella	88.5	84.5	81.8	**0.04**	0.09	0.58
Mumps	91.7	85.1	82.2	0.13	0.26	0.67
Chickenpox	89.0	87.5	84.9	0.06	0.12	0.62
Human Papillomavirus (HPV) infection	58.5	46.9	49.1	**0.01**	**0.03**	0.59
Meningitis	65.6	56.6	53.6	**0.05**	**0.03**	0.25

Statistically significant results are in **bold**.

**Table 4 vaccines-12-00473-t004:** Percentage of pregnant women who perceived the severity (quite-very severe) of the following vaccine preventable diseases.

Claims	2019–2020 %	2020–2021 %	2021–2022 %	*p*-Value 2019–2020 vs. 2020–2021	*p*-Value 2019–2020 vs. 2021–2022	*p*-Value 2020–2021 vs. 2021–2022
Hepatitis B	96.9	99.6	95.4	0.89	0.26	0.98
Poliomyelitis	95.7	98.7	95.4	0.70	0.10	0.99
*Haemophilus influenzae b* infection	66.7	78.6	72.9	0.90	0.61	0.11
Tetanus	93.8	97.4	94.0	0.94	0.46	0.06
Diphtheria	88.8	95.5	94.1	**0.01**	0.89	0.06
Pertussis	94.9	94.4	94.1	0.24	0.06	0.06
Measles	80.8	83.8	78.6	0.09	**0.003**	0.36
Rubella	70.4	81.6	76.5	0.78	0.32	0.06
Mumps	70.4	80.7	75.0	0.93	0.43	0.97
Chickenpox	69.8	73.8	68.0	0.68	0.26	0.07
Human Papilloma Virus (HPV) infection	89.3	97.3	91.2	**0.003**	0.93	0.10
Meningitis	100	99.6	98.7	**0.01**	**<0.001**	**0.03**

Statistically significant results are in **bold**.

**Table 5 vaccines-12-00473-t005:** Trust in healthcare workers and the National Health Service (NHS) before and during the COVID-19 pandemic in the 2019–2020, 2020–2021, and 2021–2022 flu seasons.

Trust in Healthcare Workers and the National Health Service (NHS) *	2019–2020 %	2020–2021 %	2021–2022 %	*p*-Value 2019–2020 vs. 2020–2021	*p*-Value 2019–2200 vs. 2021–2022	*p*-Value 2020–2021 vs. 2021–2022
I believe in the information provided by healthcare providers	98.1	94.9	96.7	0.07	0.47	0.06
NHS workers are prepared and updated on vaccinations	95.1	92.2	92.1	0.24	0.53	0.80
I have more trust in providers outside the NHS	10.9	17.9	16.2	0.93	0.78	**0.01**
NHS workers have economic interest in childhood vaccinations	11.2	7.1	6.0	0.26	**0.008**	**0.01**
NHS operators give information only about the benefits of vaccines and not the risks	33.0	23.3	25.5	**0.05**	0.06	0.44

* Percentage of women who answered “quite” or “strongly”. Statistically significant results are in **bold**.

**Table 6 vaccines-12-00473-t006:** Ranking of the information sources based on the mean scores regarding their perceived usefulness before and during the COVID-19 pandemic in the 2019–2020, 2020–2021, and 2021–2022 flu seasons.

Information Sources	2019–20 Mean * (Ranking)	2020–2021 Mean * (Ranking)		2021–2022 Mean * (Ranking)	
General Practitioner	2.0 (11)	1.97 (10)	↑	1.91 (11)	↓
Gynaecologist	3.21 (2)	2.88 (2)	=	3.03 (1)	↑
Paediatrician	2.32 (6)	2.14 (7)	↓	2.25 (7)	=
Local Health Authority/Ministry of Health information brochures	2.52 (5)	2.26 (6)	↓	2.09 (9)	↓
Vaccination clinic	2.21 (7)	2.10 (8)	↓	2.26 (6)	↑
Institutional websites	3.11 (3)	2.76 (3)	=	3.02 (2)	↑
Non-institutional websites	2.19 (8)	2.30 (5)	↑	2.28 (5)	=
Mobile applications	1.47 (13)	1.53 (13)	=	1.64 (12)	↑
Trusted physician outside the NHS	2.04 (10)	1.89 (11)	↓	2.10 (8)	↑
Antenatal classes	3.70 (1)	3.10 (1)	=	2.92 (3)	↓
Word of mouth—friends—acquaintances	2.65 (4)	2.66 (4)	=	2.45 (4)	=
Mass media (i.e., TV, radio)	2.06 (9)	1.99 (9)	=	2.03 (10)	↓
Associations against vaccinations	1.27 (14)	1.24 (14)	=	1.25 (14)	=
Other	1.65 (12)	1.60 (12)	=	1.44 (13)	↓

* Mean of perceived usefulness, measured on a scale from 1 (not useful at all) to 5 (very useful). ↑ Higher ranking than the previous year ↓ lower ranking than the previous year = equal ranking compared to the previous year.

## Data Availability

The data presented in this study are available upon reasonable request from the corresponding author.

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
