# Peer review of "Assessing the Impact of the COVID-19 Pandemic on Pregnant Women’s Attitudes towards Childhood Vaccinations: A Cross-Sectional Study"

_vaccines, 2024, doi:10.3390/vaccines12050473_

Round 1

Reviewer 1 Report

Comments and Suggestions for Authors

Vaccines 2948420

Comments to Authors

The COVID pandemic had a dramatic and lasting impact on public health and general attitudes toward science and medicine. Skepticism and hesitancy about COVID vaccinations resulted in lower vaccination rates than anticipated. Your work adds to a growing body of research that explores the broader impact on vaccine hesitancy on the greater public health.  I have made several comments that I hope you will find helpful in your current and future work in this area.

Introduction:  Many of the vaccination rates that you cite are from the United States or overall global rates. Since your study is focused on a population in a single Italian city and region, it would be more appropriate to present statistics from these areas in order to provide a better baseline for your findings.

Materials and Methods: You describe the three “waves” of the study as one prior to the pandemic (assumed to be the 1st wave), September 2020 to January 2021 (2nd wave) and October 2021 to January 2022 (5th wave).  Did you mean 3rd wave or was there 2 waves between the 2nd and 5th?

Results: You report declining participation rates from 99% (pre-COVID) to 76% (wave 2) to 47% (last wave). Do you have any insight into why there was such a difference between the three waves? Do you have any information on the non-respondents (especially in the last wave)?

I found many of the chi squared analyses (and corresponding p values) very difficult to understand and interpret. I will point out just a few findings in tables 2 through 4 that didn’t make sense to me. Table 2: Meningitis: 2019 (87.5%) compared to 2020 (87.6%) p=0.50; 2019 compared to 2021 (90.0%) p=0.03; 2020 compared to 2021 p=0.77.  The percentages for 2019 and 2020 are very close but the corresponding p values for their comparisons with 2021 are not close to being significant (0.77). Table 3: Rubella: 2019 (88.5%) compared to 2020 (84.5%) p=0.04; 2019 compared to 2021 (81.8%) p=0.09. The difference between 2019-2020 is less than that of 2019-2021 and yet the first comparison is statistically significant and the second is not. Table 4: Meningitis: the differences between the three cohorts is less than 1.3% (100%, 99.6%, 98.7%) and yet all three comparisons are statistically significant. From a practical perspective, there is no difference between the three cohorts. In fact, the three cohort percentages are closer than any other comparison.  Am I reading these tables incorrectly?

Discussion: Your results are nicely summarized and clearly presented. You raise important issues in your discussion of study limitations. It is unfortunate that you did not have enough variability in factors known to be associated with hesitancy (eg., education). Also, because your level of vaccination acceptance is so high in your sample, it is difficult to explore vaccine hesitancy. It could be helpful to compare the general level of vaccine acceptance in your sample with other studies from Italy of similar countries to see examine the  representativeness of your sample population.

Author Response

  • Introduction:  Many of the vaccination rates that you cite are from the United States or overall global rates. Since your study is focused on a population in a single Italian city and region, it would be more appropriate to present statistics from these areas in order to provide a better baseline for your findings.

Thank you for your suggestion, which enabled us to enhance the accuracy and appropriateness of our overview on vaccination coverage. Tailoring the focus to our specific geographical context, we included additional details in the manuscript (lines 64-68). Inizio modulo

  • Materials and Methods: You describe the three “waves” of the study as one prior to the pandemic (assumed to be the 1stwave), September 2020 to January 2021 (2nd wave) and October 2021 to January 2022 (5th wave).  Did you mean 3rd wave or was there 2 waves between the 2nd and 5th?

Thank you for your feedback, which allowed us to identify and correct a lack of clarity. When we talked about waves, we were referring to pandemic waves, and not to “waves” or phases of our study; we have added clarification in parentheses accordingly (lines 99-100). Thanks to your comment, we also identified an inaccuracy: according to data reported by the Istituto Superiore di Sanità (ISS) (available on the website https://www.epicentro.iss.it/ben/2022/4/diffusione-covid-19-ospedaliero-bolzano), there have been four waves of COVID-19 in Italy: from February to May 2020 (first wave), from October to December 2020 (second wave), from January in May 2021 (third wave), and from November 2021 to March 2022 (fourth wave). We have rectified this in the manuscript, replacing "5th wave" with "4th wave" (line 100), and reported the relative reference.

  • Results: You report declining participation rates from 99% (pre-COVID) to 76% (wave 2) to 47% (last wave). Do you have any insight into why there was such a difference between the three waves? Do you have any information on the non-respondents (especially in the last wave)?

Thank you for bringing this to our attention. The notable response rate during the first period was largely driven by the in-person format of the antenatal classes and the on-site distribution of paper questionnaires before the first lessons commenced. During the COVID-19 pandemic, the courses were transitioned to online platforms, and the questionnaires administered online through an access link sent via email before the course. This change may have led to reduced motivation among participants to respond promptly, potentially stemming from delayed email reading. Moreover, the pandemic undoubtedly influenced individuals' emotional well-being, leisure activities, and perception of priorities, consequently impacting their willingness to actively engage in the survey. Unfortunately, we lack information about non-respondents due to the anonymity of the questionnaire. We have included this consideration within the study's limitations (lines 391-394).

  • I found many of the chi squared analyses (and corresponding p values) very difficult to understand and interpret. I will point out just a few findings in tables 2 through 4 that didn’t make sense to me. Table 2: Meningitis: 2019 (87.5%) compared to 2020 (87.6%) p=0.50; 2019 compared to 2021 (90.0%) p=0.03; 2020 compared to 2021 p=0.77. The percentages for 2019 and 2020 are very close but the corresponding p values for their comparisons with 2021 are not close to being significant (0.77). Table 3: Rubella: 2019 (88.5%) compared to 2020 (84.5%) p=0.04; 2019 compared to 2021 (81.8%) p=0.09. The difference between 2019-2020 is less than that of 2019-2021 and yet the first comparison is statistically significant and the second is not. Table 4: Meningitis: the differences between the three cohorts is less than 1.3% (100%, 99.6%, 98.7%) and yet all three comparisons are statistically significant. From a practical perspective, there is no difference between the three cohorts. In fact, the three cohort percentages are closer than any other comparison.  Am I reading these tables incorrectly?

Thank you for your comment. As previously reported in the other results tables, the comparison was made between two years (2019-20 vs 2020-21, 2019-2020 vs 2021-2022, and 2020-2021 vs 2021-2022). Although values might look similar to each other, the significance depends on the variability of the answers, especially if only the type of replies “quite-very probable” and “quite-very severe” are considered.

  • Discussion: Your results are nicely summarized and clearly presented. You raise important issues in your discussion of study limitations. It is unfortunate that you did not have enough variability in factors known to be associated with hesitancy (eg., education). Also, because your level of vaccination acceptance is so high in your sample, it is difficult to explore vaccine hesitancy. It could be helpful to compare the general level of vaccine acceptance in your sample with other studies from Italy of similar countries to see examine the representativeness of your sample population.

Thank you for your appreciative remarks on the clear and concise presentation of our findings, as well as your acknowledgment of the significant issues we've addressed regarding the limitations of our study.  We want to assure you that we have already emphasized this aspect in our discussion section. We have meticulously integrated numerous references to compare acceptance levels of various vaccinations explored in our study with those documented in relevant literature from Italy and similar contexts. Furthermore, we specifically highlighted the imperative for additional studies to better understand how socio-economic factors influence vaccination intentions and to elucidate the specific reasons for hesitancy towards certain vaccines.

Reviewer 2 Report

Comments and Suggestions for Authors

Thank you for sharing your article assessing the impact of the COVID-19 pandemic on pregnant women's attitudes towards childhood vaccination. The following comments may help to improve the manuscript. 

L37: Does "a further 2 million deaths" also relates to children or all ages?

L39-40: Do you mean the COVID-19 pandemic when stating "the recent pandemic"? This applies throughout your manuscript. 

L47: Regarding "vaccinating pregnant patients", by which diseases are these patients affected? Please clarify.

L95-96: Please include a rationale for targeting PW from the 4th month of pregnancy. Did participants have to fulfil any further eligibility criteria such as age, nationality/language or place of residency?

L116: Regarding online forms, did you also obtain informed consent via an online platform? Could any potential participants not complete the online questionnaire due to limited internet access which in turn could have introduced some selection bias? 

L83: Did you re-assess the same group of PW during each study round or did PW differ/were selected newly during each study round? 

Table 1-5: Please check the journal's requirements how to report p-values. 

Table 6: What represent the red and green arrows and the "=" symbol? 

Comments on the Quality of English Language

The manuscript would benefit from some English editing throughout. Especially the section "discussion" contains some lengthy sentences that should be revised for better readability.

Author Response

  • L37: Does "a further 2 million deaths" also relates to children or all ages?

Thank you for your comment. According to the cited reference, the “further 2 million deaths” refer to children. For greater clarity, we have specified this better in the manuscript (line 37).

  • L39-40: Do you mean the COVID-19 pandemic when stating "the recent pandemic"? This applies throughout your manuscript. 

Thank you for your feedback, which allowed us to be more accurate on this matter. Since we always refer to the COVID-19 pandemic, we have explicitly stated this in the manuscript by replacing "recent pandemic" with "COVID-19 pandemic” (lines 39,80).

  • L47: Regarding "vaccinating pregnant patients", by which diseases are these patients affected? Please clarify.

Thank you for your comment. We acknowledge that the term "patient" in this context may be inappropriate and could lead to misunderstandings, as it might imply considering pregnancy as a pathological condition. Therefore, we removed it from the manuscript (line 47).

  • L95-96: Please include a rationale for targeting PW from the 4th month of pregnancy. Did participants have to fulfil any further eligibility criteria such as age, nationality/language or place of residency?

Thank you for this comment. We have specified in the text that, according to our corporate policy, the course begins in the fourth month of pregnancy. There are no inclusion criteria; it is open to all pregnant women receiving care at our hospital (lines 106-113).

  • L116: Regarding online forms, did you also obtain informed consent via an online platform? Could any potential participants not complete the online questionnaire due to limited internet access which in turn could have introduced some selection bias? 

Thank you for your comment, which helped us to achieve greater clarity on this matter within the text. For the second and third antenatal courses, informed consent was obtained through an online platform. We acknowledge that our manuscript may have been unclear on this point; therefore, we have added clarification at lines 116-117. As previously mentioned in the "Strengths and Limitations" paragraph (lines 397-398), the requirement for internet access for participant selection introduces a potential for selection bias.

  • L83: Did you re-assess the same group of PW during each study round or did PW differ/were selected newly during each study round? 

Thank you for this suggestion. We have emphasized in the text that these are different groups analyzed in different years (lines 95-97).

  • Table 1-5: Please check the journal's requirements how to report p-values. 

Thank you for your suggestion. Unfortunately, we couldn't find explicit instructions on this matter in the editorial guidelines. Nevertheless, we have attempted to report the p-values consistently with other articles published by the journal.

  • Table 6: What represent the red and green arrows and the "=" symbol?

Thank you for bringing this lack of clarity to our attention, we inserted a note below the table explaining their meaning.

  • The manuscript would benefit from some English editing throughout. Especially the section "discussion" contains some lengthy sentences that should be revised for better readability.

Thank you for your suggestion. We regret to acknowledge that some sections seem to lack readability, and we recognize that the quality of English writing could be improved. Nonetheless, we have endeavoured to ensure its quality by having the manuscript reviewed by a native language expert before submission. In addition, based on your input, we have reviewed the discussion section and made some minor changes to improve clarity. However, we are available should further English editing be deemed necessary.
